# Exploring Metabolic Adaptations to the Acidic Microenvironment of Osteosarcoma Cells Unveils Sphingosine 1-Phosphate as a Valuable Therapeutic Target

**DOI:** 10.3390/cancers13020311

**Published:** 2021-01-15

**Authors:** Margherita Cortini, Andrea Armirotti, Marta Columbaro, Dario Livio Longo, Gemma Di Pompo, Elena Cannas, Alessandra Maresca, Costantino Errani, Alessandra Longhi, Alberto Righi, Valerio Carelli, Nicola Baldini, Sofia Avnet

**Affiliations:** 1Biomedical Science and Technology Lab, IRCCS Istituto Ortopedico Rizzoli, 40136 Bologna, Italy; margherita.cortini@ior.it (M.C.); marta.columbaro@ior.it (M.C.); gemma.dipompo@ior.it (G.D.P.); sofia.avnet@ior.it (S.A.); 2Analytical Chemistry Lab, Fondazione Istituto Italiano di Tecnologia, 16163 Genova, Italy; andrea.armirotti@iit.it (A.A.); elena.cannas@edu.unito.it (E.C.); 3Institute of Biostructures and Bioimaging, National Research Council of Italy, 10135 Torino, Italy; dario.longo@unito.it; 4Department of Biomedical and Neuromotor Sciences, Alma Mater Studiorum-Università di Bologna, 40125 Bologna, Italy; valerio.carelli@unibo.it; 5Programma di Neurogenetica, IRCCS Istituto Delle Scienze Neurologiche di Bologna, 40139 Bologna, Italy; alessandra.maresca@isnb.it; 6Oncologic Orthopaedic Unit, IRCCS Istituto Ortopedico Rizzoli, 40136 Bologna, Italy; costantino.errani@ior.it; 7Chemotherapy Unit for Musculoskeletal Tumors, IRCCS Istituto Ortopedico Rizzoli, 40136 Bologna, Italy; alessandra.longhi@ior.it; 8Anatomy and Pathological Histology Unit, IRCCS Istituto Ortopedico Rizzoli, 40136 Bologna, Italy; alberto.righi@ior.it

**Keywords:** osteosarcoma, tumor microenvironment, acidity, sphingolipid, sphingosine 1-phosphate, lipidomics, FTY720

## Abstract

**Simple Summary:**

By studying the role of tumor acidosis in osteosarcoma, we have identified a novel lipid signaling pathway that is selectively activated in acid-induced highly metastatic cell subpopulation. Furthermore, when combined to low-serine/glycine diet, the targeting of this acid-induced lipid pathway by the FDA-approved drug FTY720 significantly impaired tumor growth. This new knowledge will provide a giant leap in the understanding of the molecular mechanisms responsible for sarcoma relapses and metastasis. Finally, we paved the way to the recognition of a novel biomarker, as our data provided evidence of significantly high circulating levels in the serum of osteosarcoma patients of S1P, a lipid member of the identified acid-driven metabolic pathway.

**Abstract:**

Acidity is a key player in cancer progression, modelling a microenvironment that prevents immune surveillance and enhances invasiveness, survival, and drug resistance. Here, we demonstrated in spheroids from osteosarcoma cell lines that the exposure to acidosis remarkably caused intracellular lipid droplets accumulation. Lipid accumulation was also detected in sarcoma tissues in close proximity to tumor area that express the acid-related biomarker LAMP2. Acid-induced lipid droplets-accumulation was not functional to a higher energetic request, but rather to cell survival. As a mechanism, we found increased levels of sphingomyelin and secretion of the sphingosine 1-phosphate, and the activation of the associated sphingolipid pathway and the non-canonical NF-ĸB pathway, respectively. Moreover, decreasing sphingosine 1-phosphate levels (S1P) by FTY720 (Fingolimod) impaired acid-induced tumor survival and migration. As a confirmation of the role of S1P in osteosarcoma, we found S1P high circulating levels (30.8 ± 2.5 nmol/mL, *n* = 17) in the serum of patients. Finally, when we treated osteosarcoma xenografts with FTY720 combined with low-serine/glycine diet, both lipid accumulation (as measured by magnetic resonance imaging) and tumor growth were greatly inhibited. For the first time, this study profiles the lipidomic rearrangement of sarcomas under acidic conditions, suggesting the use of anti-S1P strategies in combination with standard chemotherapy.

## 1. Introduction

Acidic tumor microenvironment (TME) has been included among the hallmarks of advanced cancers [1,2,3,4], as it deeply impacts on tumor behavior and generates protective conditions that, in term of local and systemic aggressiveness, increase drug resistance, survival and immunotolerance [5,6,7]. Intratumoral acidity occurs whenever cancer cells cope with a high energetic demand in a poorly perfused environment [8]. Under these microenvironmental conditions, malignant, highly glycolytic cells release waste products, including protons and lactate [9,10]. In turn, the excess of protons in the extracellular space induces a selective pressure on cancer cells that develop considerable metabolic plasticity to survive [11], with deep modifications of their metabolic profile [12,13,14]. A few reports have recently shown that, in some tumors, abnormal lipid metabolism, intracellular lipid levels, and paracrine crosstalk between lipid-secreting cells and the non-secreting counterpart are modulated by changes in TME, including hypoxia, nutrient availability and cell stress. In the context of extracellular acidity, cancer cells rewire fatty acid (FA) metabolism to counteract the suppression of glucose oxidation as a source of acetyl-CoA for the tricarboxylic acid cycle (TCA) [15,16]. More generally, abnormal lipid metabolism is fundamental for tumor growth [17,18,19], as proliferating cancer cells have an increased demand of lipids [20]. Part of these are used to form membranes, whereas some are stored as lipid droplets (LD) to prevent free FA-mediated cytotoxicity [21]. Lipids are also oxidized to acetyl-CoA to generate energy and support cancer growth [20] or act as signaling molecules to coordinate transduction cascades [17,22]. Two examples of signaling lipids are ceramide and sphingosine 1-phosphate (S1P), both generated by sphingolipid metabolism and showing opposite biological effects [23,24,25]: ceramide is pro-apoptotic and anti-tumorigenic [26], whereas S1P signaling inhibits apoptosis and stimulates cell proliferation and migration by directly activating signal transduction networks [23,27] through five distinct receptors, S1PR1-5. The S1P-activated signal transduction pathways have distinct roles in the regulation of cancer cell proliferation, migration and/or invasion in a context-dependent manner [23]. The overall effect is an increased metastatic potential. As an example, in colorectal cancer, the activation of the fatty acid synthase/S1P axis increases cancer cell proliferation, migration, formation of focal adhesions, and degradation of the extracellular matrix [28].

Here, we sought to investigate the role of acidic TME and its impact on lipid accumulation and metabolism in sarcomas. We focused on osteosarcoma (OS), an aggressive and relatively common malignancy in which an acidic TME is associated with tumor aggressiveness and resistance to antineoplastic drugs [14,15,29,30,31]. We investigated how OS cells cope and adapt to the microenvironmental stress generated by acidity by activating lipid metabolism. Since standard in vitro conditions are only partially able to recapitulate the complex interactions activated in TME [32,33], we used 3D hanging drop spheroids as a more reliable model. We demonstrated that the activation of the sphingolipid recycling pathway was able to ensure the survival of sarcoma cells under acidity,and confirmed the accumulation of LD in cancer cells also in tissue samples of sarcomas and high levels of S1P in serum samples from OS patients. Finally, by impairing the S1P-sphingomyelin pathway, we specifically and effectively targeted the subset of aggressive sarcoma cells that, under acidic microenvironmental conditions, have the potential to escape the activity of conventional antineoplastic agents. Overall, our findings suggest that impinging the sphingolipid pathway can improve acid-resistant cell disposal, with the potential for clinical translation.

## 2. Results

### 2.1. LD Accumulation in Sarcoma Tissues and Acid-Induced LD Accumulation and Migration in OS Cells

To examine the relationship between acid adaptation and LD formation in sarcomas, we first analyzed the lipid presence and distribution by Nile Red (Figure 1A and Appendix A) and Oil Red O (Figure 1B and Appendix A) staining in frozen sections of 5 OS and other 4 high-grade sarcomas (leiomyosarcoma, liposarcoma, and undifferentiated spindle cell sarcoma). Intracellular lipids were observed in all histotypes. Furthermore, LD clearly co-localized with Lamp2, an indicator of acidic environment [34], in the cellularized areas of most tissue samples (Figure 1A and Appendix A).

Cells overloaded with lipids were also observed by transmission electron microscopy (EM) (Figure 1C). We then cultured in monolayer a panel of OS cell lines under acidic (pH 6.8) microenvironmental conditions for 72 h, thereby mimicking the high extracellular proton concentration that is constantly observed in sarcomas. In all the cell lines, by Oil Red O staining, we observed intracellular accumulation of LD (Appendix A). In 143B, notably, at pH 6.8, we also noted a strong reduction in the number of cells (Appendix A). Acid-induced LD accumulation was confirmed also in hanging drop models (Figure 1D,E and Appendix A).

For the next assays, we only focused on two representative OS cell lines: 143B that are frequently used as a reproducible in vivo model [31,35,36], and HOS that are isogenic to 143B but non-metastatic [37]. Under our experimental conditions (96 h), the spheroids grown under neutral conditions increased their diameter from 200 μm to 300 μm (Appendix A), despite not developing a necrotic core (Appendix A). Conversely, the acidic stress obtained with the unbuffered medium caused the inhibition of spheroid growth (Appendix A) but did not lead to cell death, as assessed by few sparse propidium-iodide-positive cells (Appendix A) Immunostaining for the acid-related marker Lamp2 in 143B spheroids showed a diffuse positivity throughout the whole spheroid under unbuffered and thus acidic conditions. However, we noticed Lamp2 positivity in correspondence with LD accumulation in spheroids grown in neutral conditions, confirming lipid accumulation only in the presence of the development of intra-spheroid acidosis (Appendix A). Compared to neutral pH, acid-stimulated cells from spheroids showed a greater migratory ability (Appendix A) [4,38]. In conclusion, our results suggest that acidity is a considerable source of stress [31,39] that causes lipid accumulation, does not cause cell death and leads to enhanced cell migration.

Autophagy plays a key role in stress-response as well as in the regulation of lipid storage and metabolism [40] and in the survival of cancer cells under low pH conditions [41]. We therefore explored whether autophagy might be responsible for the increased lipid accumulation observed in OS cells under extracellular acidity (for a graphical representation of the autophagic regulation of LD sequestration see Figure 1F). The autophagic flux seemed to be unchanged between pH conditions (Figure 1G) but EM analysis revealed a different picture. Double-membrane organelles (indicating the key initial event of autophagy) could be clearly observed in 143B and HOS spheroids under acidic conditions (Figure 1H and Appendix A) and in sarcoma tissues (Appendix A), but not at pH 7.4 (Figure 1H and Appendix A). The average diameter of LD was 800 nm, but this value increased to 1000 nm under acidic conditions (Figure 1H and Appendix A) and these organelles marked positive for LC3 only under acidic conditions (Appendix A).

LD/LysoSensor co-localization did not increase under acidic conditions (Appendix A), whereas the time-length of the interaction between lysosomes and lipid-containing vesicles was reduced, indicating increased dynamics (Appendix A). The decreased expression of LD-associated protein PLIN2 and the increased expression of DGAT1, an enzyme that is protective against lipotoxicity, under acidic conditions (Appendix A) suggest that to avoid acid-induced lipotoxicity, a great amount of free FA is conveyed into LD [42,43].

### 2.2. Acid-Induced Lipid Overload Is Unrelated to Energy Production

To evaluate whether acid-promoted LD accumulation is used to meet an energetic need, we next investigated mitochondrial dynamics and respiration in OS cells chronically adapted to grow in acidic conditions. To this aim, we used monolayered cells, as Seahorse readouts could not be performed in 3D, and to perform the measurement of cellular respiration, cells were kept in a standard mito-stress assay buffer at standard pH value (7.4) only during the instrument readouts. We found that, although the relative expression of genes encoding mitochondrial complex I (i.e., NDUFA9 and NDUFAB1) was similar in cells grown under acidic and neutral conditions (Appendix A), a significant decrease in oxygen consumption rate (OCR), with reductions both in basal and maximal respiration, occurred only under acidity (Figure 2A,B (143B cells) and Appendix A (HOS cells)). To ascertain whether lipids accumulating under acidic conditions could be used as metabolic substrates to fuel β-oxidation, we also treated 143B cells with etomoxir, a blocker of mitochondrial acyl-CoA intake. Again, we observed a significant decrease in OCR and the relative basal and maximal respiration rates (Figure 2A,B).

FA trafficking from LD to mitochondria for energy production requires mitochondrial fusion [44,45,46,47]. On this regard, we investigated the relative expression of genes involved in mitochondrial fusion (MFN1 and MFN2) and fission (DRP1 and MFF) and found that these were significantly upregulated upon acidic conditions in 143B (Figure 2C), indicating highly dynamic organelles under acidity. Furthermore, in the same conditions, the number of mitochondria per cell was significantly increased (Figure 2D, Appendix A), whereas the average mitochondrial diameter decreased from 1500 nm to 1000 nm (Figure 2E), suggesting that low pH induced mitochondrial fragmentation rather than fusion [48,49].

Finally, lipid trafficking from the LD to the mitochondria for oxidative respiration requires mitochondria to localize near the LD. As further confirmation of the lack of energetic fueling by LD, we did not observe co-localization of these two organelles under acidic conditions, either in 143B or HOS cells (Figure 2F (143B) and Appendix A (HOS)). Intriguingly, we observed marked re-localization of mitochondrial structures from a homogenous disperse cytoplasmic distribution under neutral conditions to a centralized, perinuclear clustering in cells grown under acidity (Figure 2G and Appendix A). Lastly, the mitochondria did not show swelling or damage (Appendix A), suggesting that the lower cellular respiration in acid-grown cells was not due to mitochondrial injury.

Overall, our data suggest that acid-induced lipids accumulated in LD are not trafficked to the mitochondria and, hence, are not used as an energetic source. However, acidity may be perceived as a stress stimulus from mitochondria since it increases their dynamics, fragmentation and intracellular re-localization.

### 2.3. Intracellular Pools of Sphingomyelin and S1P Increase in OS Cells under Acidic Conditions

Given that the acid-induced LD are not primarily used by OS cells to fuel OXPHOS, we performed LC-MS/MS-based untargeted lipidomic profiling to enlighten which lipid categories are affected and differentially accumulated after OS spheroid exposure to extracellular acidity. The principal component analysis (PCA) score plot suggests distinct lipid compositions at different pH values (Figure 3A). In particular, sphingomyelin d18:1/16:0 (a sphingomyelin molecule, deriving from sphingosine (d18:1) and carrying a palmitic acid moiety (16:0) linked to the amino group, as amide)) levels were significantly increased, whereas the levels of phosphocholine 16:0/14:0 (a glycerophospholipid carrying a choline headgroup and two well defined fatty acyl chain: palmitic (16:0) and myristic (14:0) acid), which can be converted into sphingomyelin, were significantly decreased (Figure 3B), suggesting that phosphocholine is mainly converted to sphingomyelin (Figure 3C).

To verify whether sphingomyelin could be converted into S1P, we next monitored the relative expression of the kinases that phosphorylate sphingosine to form S1P: SPHK1, which encodes a cytosolic protein, and SPHK2, which encodes a nuclear isoenzyme [50]. SPHK1 was significantly upregulated in acid-treated spheroids, whereas SPHK2 was unchanged (Figure 3D). The latter observation was not surprising, as SphK2 is known to be an inducer of cellular apoptosis, whereas SphK1 stimulates growth and survival [51]. Consistent with these findings, intracellular S1P levels were significantly increased under acidity in 143B spheroids (Figure 3E), and mildly, although not significantly, also in HOS spheroids (Appendix A). Overall, our data revealed that oncogenic lipids are greatly upregulated by acidity.

### 2.4. Acid-Driven S1P Accumulation Is Responsible for Increased Tumor Cell Survival and Migration

Because sphingomyelin and sphingolipid signaling activate the pro-survival S1P-S1P receptor pathway [23], we next investigated if sphingomyelin and S1P synthesis in acid-induced OS spheroids was able to modulate cell viability. To this aim, we used myriocin, an inhibitor of serine palmitoyltransferase (SPT) [52], and its derivative FTY720 (Fingolimod), which inhibits G protein-coupled S1P receptors and blocks oncogenic signaling [53,54,55,56]. Both myriocin and FTY720 were able to impair S1P accumulation in 143B spheroids under acidity, with a greater effect shown by FTY720 (Figure 3E). Interestingly, under acidic conditions, myriocin had a weaker inhibitory effect on 143B spheroid growth than FTY720 (Figure 3F). FTY720 also greatly impaired LD formation under acidic conditions (Figure 3G, acidity and Appendix A, neutrality). By contrast, a striking reduction in cell viability to myriocin was observed when spheroids were cultured under neutral conditions (Figure 3F and Appendix A (143B) and C (HOS)), suggesting that the de novo sphingolipid synthetic pathway is prevalent under neutral but not under acidic conditions. By contrast, the results with FTY720 suggest that the sphingomyelinase pathway is prevalent under acidic conditions as a source for LD accumulation and for S1P synthesis. We next investigated how the two drugs interfere with acidity-induced cell migration [4,30,38]. In agreement with the effect on cell growth and lipid accumulation under neutral conditions, myriocin markedly decreased 143B spheroid migration that, adversely, was unaffected in response to short-term acidity. Conversely, FTY720 significantly impaired tumor cell migration under acidity but had no effect under neutral conditions (Figure 3H).

Overall, these data demonstrate that acidity induces a marked overexpression of components of the sphingolipid pathway, with specific enhancement of the sphingomyelinase pathway over the de novo pathway and S1P accumulation. Moreover, we showed that FTY720 can effectively inhibit the acid-induced S1P accumulation and reduce the survival and migration of OS cells under acidic conditions.

### 2.5. Secreted and Circulating S1P in Osteosarcoma Patients

S1P is an inflammatory mediator that can be secreted by tumor cells. In 143B, the concentration of S1P in the supernatant showed an increased trend in acid-treated spheroids compared to those growing in neutral conditions (Appendix A). Notably, S1P can also be released in the blood and its circulating levels have been correlated to the level of cancer aggressiveness in breast cancer [57]. Thus, to study the clinical relevance of S1P secretion in OS, we analyzed the S1P levels by ELISA in serum samples from OS patients at the time of diagnosis (T0) and after chemotherapy (T1). The average level of S1P at the onset was 30.8 ± 2.5 nmol/mL (*n* = 17) and was significantly higher than that detected at the end of the treatment (22.1 ± 1.9 nmol/mL, Figure 3I), suggesting this metabolite as a candidate biomarker for OS patients. The S1P value range was confirmed by targeted lipidomic analysis (*n* = 15, 30.6 ± 2.8 nmol/mL at T0 and 21.9 ± 2.1 nmol/mL at T1).

### 2.6. Acid-Induced S1P Activates the Antiapoptotic Noncanonical NF-κB Pathway

One major consequence of acidity on OS cells is the upregulation of NF-κB target genes [31], which activate the inflammatory response that promotes the survival of tumor cells [30,58]. We characterized the NF-κB pathway also in acid-treated OS spheroids. First, we confirmed increased p65 and NF-κB1 levels in nuclear extracts of 143B spheroids under acidity compared to neutral conditions (Figure 4A).

Next, we found that FTY720 significantly increased the percentage of apoptotic cells (Figure 4B), whereas myriocin was not effective, suggesting that S1P mediates the anti-apoptotic response to acidity. We also measured the cytoplasmic levels of TRAF adaptor proteins and BIRC2/3, as they have been demonstrated to form a receptor complex that leads to the activation of the non-canonical NF-ĸB pro-survival pathway [58]. Indeed, we found that under acidic conditions in OS spheroids TRAF and BIRC mRNA and protein levels were significantly increased (Figure 4C,D respectively) and FTY720 treatment decreased TRAF and BIRC protein levels. These results suggest the activation by acidosis and acid-induced S1P activity of BIRC and TRAF domains that mediate anti-apoptotic signals, possibly leading to NF-κB activation; the inhibition of S1P synthesis via FTY720 reversed such effect. Conversely, as expected, myriocin did not have any significant activity on the expression of these anti-apoptotic mediators (Figure 4C,D).

### 2.7. In OS Xenografts, Targeting Sphingolipid Metabolism Using FTY720 and Low Serine/Glycine Diet Reduces Tumor Growth

The above-reported results suggest that FTY720 treatment impairs tumor cell survival in acidic condition by inhibiting S1P-induced anti-apoptotic signaling. To verify whether these in vitro effects could be translated in vivo and to assess whether FTY720 targets and accumulates within the tumor, we used a subcutaneous xenograft model of OS. We did not evaluate myriocin in vivo because of its documented gastro- and hepatotoxicity [59]. Instead, we used dietary serine/glycine restriction, as its deprivation blocks, like myriocin, serine conversion to 3-ketosphinganine [60] and has already been used in cancer in vivo xenograft and allograft models [60,61]. Since glycine can be converted into serine and thus supports cell proliferation, the chow was depleted also of this non-essential amino acid [61]. We thus used intraperitoneal treatment with FTY720 alone or combined with serine/glycine restriction (Figure 5A).

Mice were fed normally or by the serine/glycine-free chow, starting from the time of cell inoculation. However, in contrast to the previous work of Maddocks et al. [60,61], mice did not tolerate the complete deprivation of serine/glycine, as highlighted by a significant decrease in their body weight (Appendix A). Thus, we reasoned to carry on the experiments with low serine/glycine diet by using 75% deprivation compared to levels in normal chow (0.33% vs. 1.33% serine concentration). Although decreased serine/glycine levels by 75% had no effect on body weight (Appendix A), it caused a consistent although non-significant decrease in tumor volume and a significant reduction in the fat fraction, as measured by magnetic resonant imaging (Figure 5C,D). Adversely, FTY720 treatment alone had little effect on tumor growth or fat fraction (Figure 5C,D), as its main target is the acidic cell population, which represents only a fraction of the tumor mass [62]. However, the targeting of acid-resistant cells is fundamental for the successful eradication of the disease. By contrast, the combination of the reduced serine/glycine diet and FTY720 significantly inhibited tumor growth and percentage of fat fraction at a greater extent than low serine diet alone (Figure 5C).

To confirm that intraperitoneal FTY720 treatment successfully targeted the tumor and that the observed growth-inhibition was specifically due to the FTY720 effect at the tumor site, we performed untargeted lipid profiling of the extracted tumors from FTY720- and vehicle-treated mice. A signal at 308.25 m/z corresponding to FTY720 was clearly detected in the tumor and only in the treatment group (Appendix A). Furthermore, the FTY720 signal represents the most important feature in the variable importance in projection (VIP) plot generated from the partial least squares discriminant analysis (PLS-DA) of the untargeted lipidomics dataset (Appendix A). From these plots, it was also evident that the two groups have significantly different lipidomic profiles. In particular, as shown in Table 1 and Appendix A, the single treatment with FTY720 significantly affected the sphingolipid pathway, providing additional evidence that this drug has the potential to be used as anticancer treatment to target the downstream pro-survival effects induced by intratumoral acidity and mediated by lipid accumulation and sphingolipid pathway activation in OS cells.

## 3. Discussion

The mechanisms that promote cancer aggressiveness and persistence under acidic conditions are largely unknown. Tumor-specific metabolic adaptations have been profiled in short-term acid-treated cells, highlighting a meaningful reprogramming of several metabolic pathways [14], including lipid metabolism [12,13,15,42]. Here, the role of lipids in sarcomas was investigated by a bioenergetic and signaling point of view.

We used OS spheroids to mimic the OS microenvironment under 3D conditions. As previously demonstrated in monolayers [30], extracellular acidity inhibited cell growth and increased their migration. We also observed a striking accumulation of LD both in tissue samples and in acid-exposed OS cells. It has been previously reported that NMR visible mobile lipids may accumulate in LD in the case of cancer cell growth arrest [63]. It is thus possible also in our cell model that LD accumulation is due to the acid-induced growth arrest. Although this possibility hasn’t been explored, it might be occurring that OS cells take advantage of lipid accumulation and undergo an adaptation process to develop strategies to cope with survival, growth arrest and acidic stress.

The presence of LD is deeply connected with autophagy. LD can be produced as a result of autophagy or are used to support the autophagic process [64]. We found that, under acidic stress, OS cells activate a specific, lipid-dependent form of autophagy, i.e., lipophagy, that is partially responsible for the sequestration of LD to form autolipophagosomes [40,65].

Regarding the role of LD accumulation in supporting the cancer cell energetic needs, in contrast to the current literature [15], we were unable to demonstrate, at a low extracellular pH, a role for LD as an energy source for the electron transport chain in mitochondria. Accordingly, no gross morphological changes were observed in the mitochondrial ultrastructure. On the contrary, we observed an increased number of mitochondria, a reduced mitochondrial diameter, the induction of expression of MFN1, MFN2 and DRP1 genes, and the re-localization of mitochondria at the perinuclear region, all features of mitochondrial fragmentation [48,49]. Recent reports have highlighted that fragmented mitochondria in fusion-deficient Mfn1-knockout cells do not efficiently metabolize FA [47], eventually resulting in increased storage of FA within LD [47]. Taken together, it is reasonable to conclude that, also in acid-stressed OS, mitochondrial fragmentation reduces the rate of β-oxidation since cells possibly strive to compensate for an impaired fusion by increasing organelle number and mRNA expression of MFN1 and MFN2. At the same time, accumulation of LD may prevent cytoplasmic toxic lipid overload and accumulation of acylcarnitines, which would otherwise directly disrupt mitochondrial integrity [43]. Furthermore, re-localization of mitochondria under acidic conditions may suggest an alert state that precedes apoptosis, as already demonstrated in neurons and melanoma cells [66,67], or the activation of a mitochondrial-to-nucleus communication pathway, as described for GPS2, supporting basal mitochondrial biogenesis [68]. Together, these data indicate that acid-induced lipids accumulate in LD, but do not convey to the mitochondria and, hence, are not used to support tumor cell respiration under acidity.

Through lipidomic analysis, we enlightened a deep reprogramming of the sphingolipid pathway at low pH, overexpression of sphingomyelin and release of S1P by acid-cultured OS spheroids. We detected S1P also systemically, in the serum of OS patients, that was possibly secreted by tumor cells since its levels were significantly decrease after the completion of the therapeutic protocol. Plasma S1P levels have been found almost twice in ovarian cancer patients than in healthy controls [69] and are prognostic in lung cancer [70]. Secreted circulating S1P increases metastasis [71,72,73,74] and has thus the potential of being validated as a prognostic biomarker [75,76]. For the future, a larger case-control prospective study may also validate S1P as a valuable biomarker in OS.

S1P accumulation is known to activate oncogenic signaling [23,27] and, through an autocrine circuit, it inhibits apoptosis, induces cell proliferation and/or migration and increases drug resistance [23]. Accordingly, also in OS cells, we indirectly found that S1P leads to the acquisition of a pro-migratory and pro-survival phenotype of the acid-adapted tumor cell subpopulation.

Subsequently, to confirm the role of S1P in OS cells surviving in an acidic TME, we used FTY720, a modulator of the sphingomyelin pathway and of S1P receptor. FTY720 is FDA approved for the treatment of multiple sclerosis, and it is currently under trial for the treatment of breast carcinoma, glioblastoma and anaplastic astrocytoma (NCT03941743 and NCT02490930). FTY720 has several anticancer effects [77,78,79,80,81] but its activity in OS has not been explored so far. Despite being considered an S1P receptor modulator, FTY720 is also able to induce changes in the enzymes of the sphingolipid pathway, including SPHK1 [54,55,56], thus providing an explanation for the reduction of S1P seen in our model. FTY720 inhibited LD and S1P accumulation in acid-treated spheroids, and, as a consequence, impaired cell survival and migration. This effect was mediated by the impairment of the non-canonical NF-ĸB pathway that, in acidity, we already demonstrated to regulate OS stemness and survival [30,82] through the involvement of TRAF adaptor proteins, and BIRC2 and/or BIRC3 [30,31,58]. Indeed, we found that FTY720 treatment in acid-stressed cells blocked the expression of TRAF and BIRC2/3, possibly by blocking S1P. SPHK1 has been already linked by previous reports to NF-κB-dependent IL-6 secretion, with persistent activation of STAT3 and the consequent upregulation of S1P receptor 1 [83].

Since also the sphingolipid de novo pathway contributes to S1P regulation, in addition we tested the effects of myriocin [84,85]. Myriocin treatment dramatically decreased OS cell survival only under neutral but acid conditions. One possible explanation is that at neutral pH, glycolytic carbons is shunted to serine and glycine synthesis through phosphoglycerate dehydrogenase (PHGDH) [86], and, in this case, lipids and ultimately S1P are formed via PHGDH activity. On the opposite side, under acidity, the lower glycolytic flux might impair lipid synthesis from serine and glycine by reducing the availability of glycolytic carbons. As a compensatory mechanism, cells activate S1P synthesis via the sphingomyelinase recycling pathway.

Based on our in vitro findings, we then aimed at affecting OS growth by targeting the sphingolipid metabolism in a mouse xenograft model. We sought to administer FTY720 in combination with a low serine/glycine diet in order to impair both the de novo synthesis and the recycling pathway that, according to in vitro data, are activated, respectively, by the neutral and acidic area of the tumor microenvironment. Myriocin had to be replaced by dietary serine/glycine restriction due to its remarkable adverse effects [59]. Notably, the nonessential amino acids serine and glycine are substrates in multiple anabolic processes that support cancer cell growth and proliferation. While some cancer cells upregulate the de novo serine synthesis, many others rely on exogenous serine for optimal growth [60,61]. In OS xenografts, combining FTY720 to a serine/glycine restrictive diet significantly affected the sphingomyelin pathway, tumor growth and fat fraction. Serine/glycine restriction alone showed an effect, although at a lower extent, suggesting that the inhibition of sphingolipid pathway should be considered for future therapies, possibly combining strategies that can affect, both the sphingomyelinase recycling pathway and the de novo synthesis pathway.

## 4. Materials and Methods

### 4.1. Cell Lines and Sphere Cultures

Unless otherwise stated, all reagents were purchased from Sigma-Aldrich, St. Louis, MO, USA). 143B, HOS, MG-63 and Saos-2 cell lines were purchased from the American Type Culture Collection (ATCC) and cultured in IMDM (Life Technologies, Carlsbad, CA, USA) plus penicillin (20 U/mL), streptomycin (100 mg/mL), and 10% heat-inactivated fetal bovine serum for a range of 10–20 passages from thawing. Cells were maintained at 37 °C in humidified atmosphere with 5% CO_2_. All cell lines were tested against mycoplasma with MycoAlert Mycoplasm Detection Kit (Lonza, Basil, Switzerland) in February 2019. All cell lines were validated (LGC Standards, Milan, Italy) in 2014. For the 2D experiments, to obtain pH 7.0 and pH 6.8 medium, IMDM was supplemented with sodium bicarbonate in the amount calculated by the Henderson-Hasselbach equation for each pH value. Experiments in 2D were performed under acute or chronic acidosis. In the case of acute exposure, after seeding, cells were incubated for 72 h with 7.4 or 6.8 pH. To obtain cells chronically adapted to acidosis, 143B and HOS cells normally cultured in IMDM complete medium were transferred to pH 7.0 for a week. As soon as the cells recovered from pH 7.0 and started growing at a rate comparable with that of pH 7.4, cultures were transferred to IMDM at pH 6.8. The cells needed about 8 weeks to recover from the low pH.

For 3D models, to generate neutral-grown spheroids, RPMI was added with sodium bicarbonate and adjusted to pH 7.4, whereas, to obtain acidic-grown spheroids, the medium was left unbuffered without the supplementation of sodium bicarbonate. This allowed the spheroids to adjust the pH according to their density and growth. For all the experiments performed in 3D, spheroids were thus grown for 96 h at pH 7.4 or unbuffered medium. The medium pH in unbuffered condition was measured and remained stable over time between 6.6 and 6.7. To obtain 3D spheroids by the hanging-drop method, 5 × 10^3^ cells were plated in 96-well Round Bottom Ultra-low attachment plate (Costar, WA, USA) in 200 mL of filtered pH 7.4 or unbuffered RPMI. Additional 200 mL of medium were added to each well and the lid placed over the plate, with specific supports fixed at every corner. The plate was flipped and incubated in gentle shaking at 37 °C and 5% CO_2_ overnight. The following day, the plate was flipped again, 200 mL of medium were removed and the time point was indicated as T0; the spheroids were then let grow for additional 5 days. When indicated, myriocin or FTY720 were added at T0 (0.75 μm myriocin, 0.15 μm FTY720) Spheroids were let grown up to 96 h.

For cell growth curves, hanging drop spheroids were formed as described. Drugs were added at the IC50 at T0 and spheroids were trypsinized and counted every 24 h.

### 4.2. Immunofluorescence 

For immunofluorescence, frozen tumor tissues were included in OCT (TissueTek, Alphen aan den Rijn, The Netherlands) and fixed with 3.7% paraformaldehyde for 20 min. Tissues samples were incubated with anti-Lamp2 (1:400 Sigma-Aldrich, #HPA029100), Nile Red (1:1000, Sigma Aldrich), followed by secondary anti-rabbit Alexa fluor 647 (1:250, Life Technologies). Nuclei were counterstained with 2.25 µg/mL of Hoechst 33258. For the analysis of in vitro models, 143B or HOS cells spheroids were centrifuged at 8000 rpm for 8 min with Cyto-Tek centrifuge (Electron Microscopy Science, Hatfield, PA, USA) and fixed with 3.7% paraformaldehyde 20 min or ice-cold methanol for 10 min, depending on the specific antibody datasheet. Cells were incubated with anti-LC3 (1:200, Cell Signaling, #2775s, Leiden, The Netherlands), Nile Red (1:1000, Sigma Aldrich), lysosensor yellow/blue (1:1000, Life Technologies), anti-Lamp2 (1:400 Sigma-Aldrich, #HPA029100), anti-surface of intact mitochondria (1:50, Millipore, #MAB1273, Burlington, MA, USA), anti-nucleoporin (1:50, Sigma Aldrich, #HPA019661), followed by secondary anti-mouse FITC conjugate (1:1000, Chemicon, Burlington, MA, USA), anti-rabbit Alexa fluor 488 (1:500, Life Technologies) or anti-mouse Alexa fluor 647 (1:250, Life Technologies). For time-lapse microscopy, 143B cells were stained with Nile Red (1:1000, Sigma Aldrich) or Lysosensor (1:1000, Life Technologies). Frames were acquired every 60 s for 8 min. Images were acquired by confocal microscopy (Nikon, TI-E). Live/dead staining was performed according to the manufacturer’s instructions (Life Technologies). Spheroid diameter was measured on live/dead images with Fiji software (NIH, Rockville, MD, USA).

### 4.3. Time-Lapse Immunofluorescence

For time-lapse microscopy, 143B cells were stained with Nile Red (1:1000, Sigma Aldrich) or Lysosensor (1:1000, Life Technologies). Frames were acquired every 60 s for 8 min. Images were acquired by confocal microscopy (Nikon, TI-E, Tokyo, Japan).

### 4.4. Oil Red O Staining 

Cells grown on coverslips were fixed with 3.7% (wt/vol) paraformaldehyde (PFA) for 10 min before staining with 5 mg/mL Oil Red O solution in 60% (vol/vol) isopropanol for 10 min. After nuclear counterstain with hematoxylin, bright-field images were acquired at ×40 magnification using a DS-Ri2 (Nikon) and lipid droplet area per cell was evaluated with Fiji software.

### 4.5. Transmission Electron Microscopy

Osteosarcoma tissue samples and 143B and HOS cells were fixed with 2.5% glutaraldehyde 0.1 M cacodylate buffer pH 7.6 for 1 h at room temperature. Post-fixation, samples were transferred in 1% OsO4 in cacodylate buffer for 1 h, and then were dehydrated in an ethanol series and embedded in Epon resin. Ultrathin sections stained with uranyl acetate and lead citrate were observed with a Jem-1011 transmission electron microscope (Jeol Inc, Peabody, MA, USA). For the in vitro models, at least 200 cells were examined for each sample.

### 4.6. Western Blotting

143B hanging-drop spheroids were harvested in hot lysis buffer (SDS 10% and TRIS HCl 0.5 M pH 6.8) containing protease and phosphatase inhibitors and centrifuged at 13,000× *g* for 10 min at 4 °C to remove insoluble debris. Protein concentrations were analyzed using BCA Protein Assay Kit (Thermo Fisher Scientific, Waltham, MA, USA). 30 μg of proteins were loaded and separated by SDS-PAGE and transferred to a 0.2 μm nitrocellulose membrane (Thermo Fisher Scientific). After blocking for 1 h in PBS-Tween 0.5% with 5% non-fat dry milk, membranes were incubated with the primary antibody for 1 h or overnight, washed and incubated for 1 h with horseradish peroxidase-conjugated secondary antibodies (Amersham, Buckinghamshire, UK). The membranes were washed and incubated with an enhanced chemi-luminescence substrate (ECL; Thermo Fisher Scientific). The antibodies were as follow: anti-LC3 1:1000 (Cell Signaling, #2775s); anti-p62 1:1000 (BD, #610833); anti TBP 1:1000 (Santa Cruz, #sc-204, Santa Cruz, CA, USA). The signal from each band was quantified by dedicated software for densitometric evaluation (ImageJ, NIH, Rockville, MD, USA).

### 4.7. Immunoelectron Microscopy

143B cells were fixed with 1% glutaraldehyde 0.1 M cacodylate buffer for 1 h at 4 °C. Samples were dehydrated in an ethanol series and embedded in EPON resin. Thin sections were pre-incubated with 5% normal goat serum, incubated with anti–LC3 antibody (Cell Signaling, #2775s) 1:10 in TBS overnight at 4 °C, and incubated with goat anti-rabbit antibody conjugated with 10 nm colloidal gold, diluted 1:10 in TBS, for 1 h at room temperature; some grids underwent colloidal gold amplification with a Silver Enhancer kit. Negative controls were samples not incubated with the primary antibody. Grids containing ultrathin sections were stained with uranyl-acetate and lead citrate and observed with a Jeol Jem-1011 transmission electron microscope at 100 kV. Mitochondria number in 143B cell spheroids cultured for 96 h were counted from EM images with 2 μm scale bar. For morphometric analysis, micrographs were taken at ×15,000 magnification and short axis (diameter) were determined by using the iTEM software (Olympus Soft Imaging Solutions, Munster, Germany).

### 4.8. RNA Isolation and Gene Expression

RNA was extracted with NucleoSpin RNA II (Macherey-nagel, Oensingen, Switzerland) and the retrotranscription was performed with MuLV reverse transcriptase (Applied Biosystems, Foster City, CA, USA). Real-Time polymerization chain reaction (Real-time PCR) was performed by amplifying 500 ng of cDNA using the SsoAdvanced Universal Probe Supermix (Biorad, Hercules, CA, USA) and the CFX96Touch instrument (Biorad). Probes and primers were selected using a web-based assay design software (www.universalprobefinder.com, Roche). For specific primer sequences see Appendix A. Since under acidic conditions no universal housekeeping is suitable for normalization (92), four different housekeeping were used: Gusb, YWHZ, 18S RNA, GAPDH.

### 4.9. Measurement of Oxygen Consumption Rate

The oxygen consumption rate (OCR), indicative of mitochondrial respiration, was measured in live cells, using the extracellular flux analyzer Seahorse XF24 instrument (Agilent Technologies, Santa Clara, CA, USA). Each cell line and condition was seeded in 5 wells of an XF 24-well cell culture microplate (Agilent Technologies) at a density of 60,000 cells/well in 250 μL of DMEM and incubated for 24 h at 37 °C under a 5% CO_2_ atmosphere. The growth medium was then replaced with 575 μL of prewarmed bicarbonate-free RPMI serum free medium supplemented with 1 mM pyruvate, 2 mM glutamine, and 10 mM glucose. Neutral or acidic-grown cells were both incubated with pH 7.35 medium at 37 °C for 1 h before starting the assay. After baseline measurements, OCR was measured after sequentially addition of oligomycin (1 μΜ), FCCP (0.5 μM), rotenone (1 μM) and antimycin A (1 μM). OCR values were normalized to protein content measured by Sulforhodamine (SRB) assay, following a standard protocol. For data analysis, the following parameters were evaluated [87]: basal respiration, maximal respiration. For measurements of endogenous β-oxidation, cells were incubated in fatty acid oxidation assay buffer (111 mM NaCl, 4.7 mM KCl, 1.25 mM CaCl_2_, 2 mM MgSO_4_, 1.2 mM NaH_2_PO_4_, 2.5 mM glucose, 0.5 mM carnitine, 5 mM HEPES) and added of etomoxir 40 μM for 15 min prior to addition of oligomycin, FCCP, rotenone and antimycin.

### 4.10. Migration Assay

For migration assays, transwells with uncoated permeable support and 8 μm pores were used. 3 × 10^4^ viable cells, obtained from trypsinized spheroids, were seeded in the top compartment of a Boyden chamber and incubated in 5% CO_2_ at 37 °C for 6 h. The bottom compartment was added of pH 7.4 or 6.8 medium. After 6 h, cells attached to the upper surface of the filter were mechanically removed by scrubbing with cotton swabs. Chambers were stained with 0.5% crystal violet diluted in 100% methanol for 30 min, rinsed in water and examined under bright-field microscopy. Values for migration were obtained by counting 5 fields per membrane (20× objective) and represent the average of three independent experiments.

### 4.11. Lipidomics

Spheroid samples were extracted by adding 1 mL of isopropanol and sonicating at room temperature for 10 min. This simple two-phase extraction protocol has already been proved to be effective for untargeted lipidomics experiments [88]. Samples were then centrifuged at 9000× *g* for 10 min and 0.9 mL of supernatant were transferred to glass vials and dried under nitrogen stream. When performed on tissues samples, the tumors were dismembered and the powder weighted, in order to normalize the amount of dissolving solution. The day of analysis, the dried lipid content was then re-dissolved in 0.1 mL of a 9:1 methanol:chloroform solution and analyzed by liquid chromatography coupled to high-resolution mass spectrometry. Five individual biological replicates of spheroids were prepared for the two experimental groups, whereas the tumor samples analyzed were *n* = 7 for each group. Untargeted lipidomics to analyse spheroids was performed on a UPLC Acquity system coupled to a Synapt G2 QToF high-resolution mass spectrometer. Lipids were separated on a CSH C18 column (1.7 μm particle size, 2.1 × 100 mm). Mobile phase A consisted of acetonitrile:water (60:40) with 10 mM ammonium formate and mobile phase B consisted of acetonitrile:isopropanol (10:90) with 10 mM ammonium formate. The following gradient program was applied: 15% B for 1 min after injection, then increase to 60% B in 9 min, then to 75% B in 8 min and then to 100% B in further 2.5 min. An isocratic 100% B step was then maintained for 2.5 min and the column was subsequently reconditioned to 15% B for 2 min. Total run time was 25 min with the following conditions: flow rate, 0.4 mL/min; column temperature, 55 °C; injection volume, 6 μL. The instrument was operated in positive ESI mode. MS source parameters were as follows: capillary and cone voltages were set at 2.8 kV and 30 V respectively. Source and desolvation temperature were set at 100 °C and 450 °C respectively. Desolvation gas and cone gas (N2) flows were 800 and 50 L/h, respectively. Mass spectra were recorded in MSe mode with MS/MS fragmentation performed in the trap region on the instrument. Low-energy scans were acquired at a fixed 4 eV collision energy, and high-energy scans were acquired using a collision energy ramp from 20 to 40 eV. Spectra were recorded at a mass resolution of 20,000 in the range of m/z 50 to m/z 1200. The scan rate was set to 0.3 spectra per second. A leucine-enkephalin solution (2 ng/mL) was continuously infused in the ESI source (4 μL/min) and acquired every 30 s for real-time mass axis recalibration. All the samples were run in random order. For data analysis we applied multivariate data analysis of LC-MS features by using the Waters Markerlynx 4.1 software with the EZinfo 2.0 (Umetrics, Umea, Sweden) software package. Principal Component Analysis (PCA) [89] was performed. The peak areas of the observed features were normalized by the total ion current and Pareto-scaled prior to PCA. Scores plots were used to visualize differences between the two different spheroid growing conditions (acidity and normal pH) [90]. Lipid identification was performed by interrogating the web-based algorithms METLIN [91] and LIPIDMAPS [92] using the accurate mass measured for each feature. The following adduct species were searched: [M + H]^+^, [M + NH^4^]^+^, [M + H − H_2_O]^+^, [M + Na]^+^, and [M + K]^+^. A maximum of 5 ppm tolerance in mass accuracy was allowed. Putative IDs were finally verified using class-specific retention time and tandem mass analysis [93].

The same serum samples analysed for S1P ELISA were also analysed for targeted following a targeted UPLC-MS/MS analysis, with a previously published protocol [94].

### 4.12. ELISA 

S1P was measured in cell lysates or human serum samples with the Sphingosine 1-Phosphate Assay Kit (Echelon, San Jose, CA, USA) according to the manufacturer’s instructions. S1P in cell lysates was normalized on total amount of protein, measured by BCA Protein Assay (Pierce, Waltham, MA, USA).

Traf 1/2 and Birc 2/3 were measured in spheroid lysates with human baculoviral IAP repeat-containing protein 2/3 ELISA Kit (MYBiosource, San Diego, CA, USA) or human tumor necrosis factor-associated factor 1 ELISA KIT (MYBiosource) according to manufacturer’s instructions. Again, Traf1/2 or Birc 2/3 in lysates were normalized on total amount of protein, as assessed by BCA Protein Assay (Pierce).

To assess nuclear translocation of NF-kB components, nuclear protein extracts were obtained and quantified with a Nuclear Extraction Kit (Cayman Chemical, Ann Arbor, MI, USA). Equal amounts of protein lysates were then used for NF-kB Transcription Factor Assay Kit quantification (TransAM, ActiveMotif, Carlsbad, CA, USA) according to manufacturer’s instructions.

### 4.13. Pyknotic Nuclei

Spheroids were grown for 5 days as indicated; at T4, cells were collected and cytocentrifuged at 800 rpm for 5 min on glass coverslip and next fixed in 3.7% paraformaldehyde 20 min. Nuclei were stained with 2.25 µg/mL of Hoechst 33258. Cells with apoptotic nuclear bodies were counted in 12 different fields on 6 different glass coverslips/condition by using a 20X objective. Results were expressed as percentage of apoptotic cells over the field.

### 4.14. In Vivo Animal Study

Balb/c nude mice were housed and maintained in a pathogen-free environment. All the procedures involving the animals were conducted according to the national and international laws on experimental animal (L.D. 26/2014; Directive 2010/63/EU) and to the approved experimental protocol procedure (Authorization N° 229/2016-PR). No validated non-animal alternatives are known to meet the objectives of the study. 143B human cells (1 × 10^6^) were subcutaneously injected with the reduced growth factor matrigel (BD Bioscience, Erembodegem, Belgium) in the flank of 5 weeks old male mice (Charles River Laboratories International). Two sets of experimental diets were used, both based on Baker Purified Amino Acid Diet from TestDiet (St. Louis, MO, USA): control diet contained all essential amino acids plus serine, glycine, glutamine, arginine, cystine, and tyrosine; serine and glycine-free diet was the same as the control diet, but without serine and glycine (or present in the indicated percentage), with the other amino acid levels increased proportionally to achieve the same total amino acid content. Details of each diet have been previously described [60,61]. Serine-glycine free diet and control diet were available from the day of tumor injection. Four days after the inoculation, mice were randomly separated into groups and assigned to pharmacological treatments (for each group *n* = 7). FTY720 was dissolved in NaCl 0.9% to reach a final concentration of 2 mg/mL and injected intraperitoneally (10 mg/kg). Groups were administered as follows: group 1) i.p. saline solution + control diet; group 2) FTY720 10 mg/kg + control diet; group 3) Serine-glycine free diet + i.p. saline solution; group 4) FTY720 10 mg/kg + serine-glycine free diet. Treatments were administered on a daily basis for 19 days. Mice were monitored for mortality daily, while weight and tumor volumes were recorded three times per week. Animals considered to be in pain, distress or in moribund condition were examined by the staff veterinarian or authorized personnel and, if necessary, humanely sacrificed to minimize undue pain or suffering. Mice were sacrificed 19 days after tumor inoculation with isoflurane gas 0.5% (O_2_ 95%). Tumor volume was calculated according to the following formula: Volume = (W^2^ × L)/2 where W refers to the maximum width and L refers to the maximum length. Tumor growth was expressed as volume (mm^3^). For lipidomic analysis, tumor xenografts were frozen, dismembered and the weights of each tumor was measured. 

### 4.15. Patient Information and Data Collection

For the data presented on human tissues, patients diagnosed with osteosarcoma, leiomyosarcoma, liposarcoma or spindle cell sarcoma were surgically treated at the Istituto Ortopedico Rizzoli. Written informed consent was obtained and the study was approved by ethical committee number 0020204 of 31/7/2009 and 0033626 of 9/11/2011. For the data presented in Figure 4I, 17 patients were diagnosed with non-metastatic osteosarcoma and serum was obtained at the time of diagnosis (T0). Patients were then assigned to chemotherapy and the serum was collected again at the end of treatment (T1), in a disease-free setting. T1 was, in most cases, collected 12 months after T0. All patients came from Istituto Ortopedico Rizzoli. Written informed consent was obtained and the study was approved by the ethical committee number 0000184 of 1/7/2015.

### 4.16. Statistical Analyses

Quantitative results were expressed as arithmetic mean plus or minus the standard error of the mean (SEM), from at least three independent experiments. Data were considered as non-parametric. Mann-Whitney test or two-tailed Student’s t-test and *p* < 0.05 was considered statistically significant. For S1P detection in human serum samples, paired two-tailed Student’s *t*-test was performed and *p* < 0.05 was considered statistically significant. Bonferroni two-way ANOVA was used for statistical analysis to compare the average tumor volume among groups. Results of quantitative data are expressed as mean ± S.E.M. All the statistical analysis was performed using Prism6 software (Graph-Pad, San Diego, CA, USA).

## 5. Conclusions

In conclusion, our results demonstrate that, in OS, the more aggressive subset of tumor cells that survive under acidity depends on sphingolipids, indicating S1P as a promising therapeutic target for these elements that would otherwise escape conventional treatments. We provide a new avenue for the development of strategies interfering with LD accumulation in order to prevent or target cell subpopulations of the acidic compartment. As serine uptake supports the Warburg effect [58], the therapeutic utility of the S1P inhibitor FTY720 could be supplemented by dietary restriction of serine/glycine. Although serine and glycine are found almost ubiquitously in food, our data demonstrate that it is possible to reduce tumor growth by partial removal of these aminoacids from the diet. Alternatively, FTY720 could be used as a supplement to conventional chemotherapy. In the latter case, FTY720 would impinge on the acid-resistant tumor subpopulation that normally resists anti-proliferative strategies [30,38]. Altogether, our study provides a fundamental leap in the knowledge of cancers that form a protective acid microenvironment and provides insights likely to be used for novel therapeutic approaches against sarcomas.

## Figures and Tables

**Figure 1 cancers-13-00311-f001:**
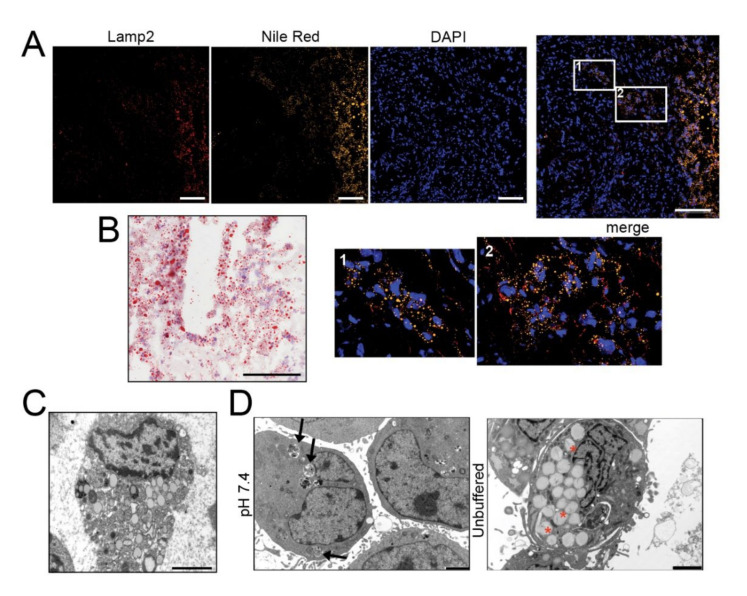
The acidic microenvironment promotes intracellular lipid accumulation in OS cells. (**A**–**C**) Tumor tissues from OS patients stained with Nile Red (for LD, yellow), the acidic marker Lamp2 (red), nuclear staining with Hoechst (blue) (**A**), and with Oil Red O for bright field visualization (counterstaining with hematoxylin) (**B**), or visualized by EM (**C**); scale bars: Nile Red O and Lamp2, 50 μm; Oil Red O, 100 μm; EM, 2 μm; (**D**) EM of 143B spheroids cultured for 96 h in neutral or unbuffered medium. Arrowheads indicate autophagosomes, whereas asterisks indicate membrane-enclosed LD. Scale bar: 2 μm; (**E**) Oil Red O staining of 143B spheroids cultured for 96 hrs in neutral or acidic medium (counterstained with hematoxylin). Scale bar: 100 μm. Right panel shows quantification of Oil Red O images. Data presented as mean ± S.E.M of *n* = 10 different fields from 3 independent experiments (unpaired two-tailed Mann-Whitney test; *** *p* < 0.001); (**F**) Schematic representation of the autophagic process: free FA are conveyed into LD from the ER, a process that is regulated by DGAT1 activity (I); autophagosome fuse with LD to form LC3-II-expressing autolipophagosome with double membranes (II); autolipophagosome fuse with lysosomes to form autolipophagolysosome (III); (**G**) Immunoblots of homogenates of 143B spheroids cultured for 5 days. Cells were treated with a saturating dose (50 nM) of bafilomycin A1, during the last 4 h of incubation to improve the detection of autophagic components. The protein expression levels of LC3-II and p62 were quantified with ImageJ software analysis and normalized to those of the housekeeping gene TBP. *n* = 3 independent experiments; (**H**) EM analysis of LD in 143B spheroids. The graph shows the percentage of cells containing double-membrane (dm) droplets relative to the number of cells containing LD. Scale bar: 500 nm; Data presented as mean ± S.E.M of *n* = 7 different fields from 3 independent experiments (unpaired two-tailed Mann-Whitney test; ** *p* < 0.01).

**Figure 2 cancers-13-00311-f002:**
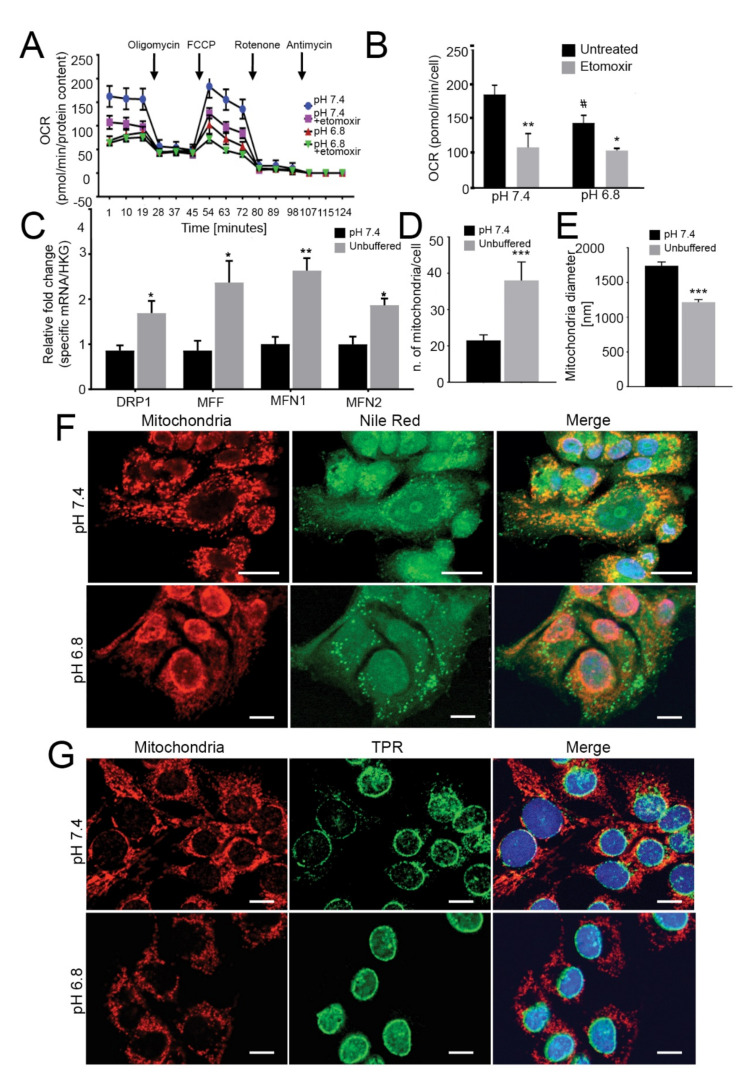
Mitochondria have low oxygen consumption and β-oxidation rates and relocalize to perinuclear regions under acidity. (**A**) OCR of monolayer 143B tumor cells chronically adapted to acidosis expressed as pmol O_2_/min normalized to protein content under basal conditions and after the injection of oligomycin, carbonyl cyanide 4-(trifluoromethoxy) phenylhydrazone (FCCP), rotenone or antimycin A. Graph shows OCR after antimycin A subtraction, to ensure removal of non-mitochondrial oxygen consumption. To evaluate β-oxidation of intracellular pool of fatty acids, 40 μM etomoxir was added to the cells 15 min prior to measuring the OCR. The graph shows a representative experiment; (**B**) Graph shows basal respiration in the presence or absence of etomoxir. Data are presented as the mean ± S.E.M. of *n* = 12 from 3 independent experiments; unpaired two-tailed Mann-Whitney test; * *p* < 0.05, ** *p* < 0.01 for etomoxir vs untreated, # *p* < 0.01 for untreated acid vs untreated neutral; (**C**) Real Time PCR analysis of the indicated genes normalized to four housekeeping genes (HKG; GAPDH, GUSB, RNA18S, and YWAHZ) in 143B spheroids cultured for 96 h Data presented as mean ± S.E.M. of *n* = 6 RNA samples from 3 independent experiments; unpaired two-tailed Mann-Whitney test; * *p* < 0.05, ** *p* < 0.01 for acid vs neutral; (**D**) Number of mitochondria/cell. Mitochondria in 143B cell spheroids cultured for 96 h were counted from EM images with 2 μm scale bar. Data presented as mean ± S.E.M. of *n* = 7 different fields from 3 independent experiments; unpaired two-tailed Mann-Whitney test; *** *p* < 0.001; (**E**) Mitochondrial diameter was measured from EM images with 2 μm scale bar in 143B cell spheroids cultured for 96 h Data presented as mean ± S.E.M. of *n* = 9 different fields from 3 independent experiments; unpaired two-tailed Mann-Whitney test; *** *p* < 0.001; (**F**) Immunostaining of mitochondria (anti-surface mitochondria, red) with LD (Nile Red, green) in monolayer 143B tumor cells chronically adapted to acidosis. Nuclear staining with Hoechst (blue). Scale bar: 20 μm; (**G**) Immunostaining of monolayer 143B tumor cells chronically adapted to acidosis: staining of mitochondria (anti-surface mitochondria, red) with the nucleoprotein TPR (green). Nuclear staining with Hoechst (blue). Scale bar: 20 μm.

**Figure 3 cancers-13-00311-f003:**
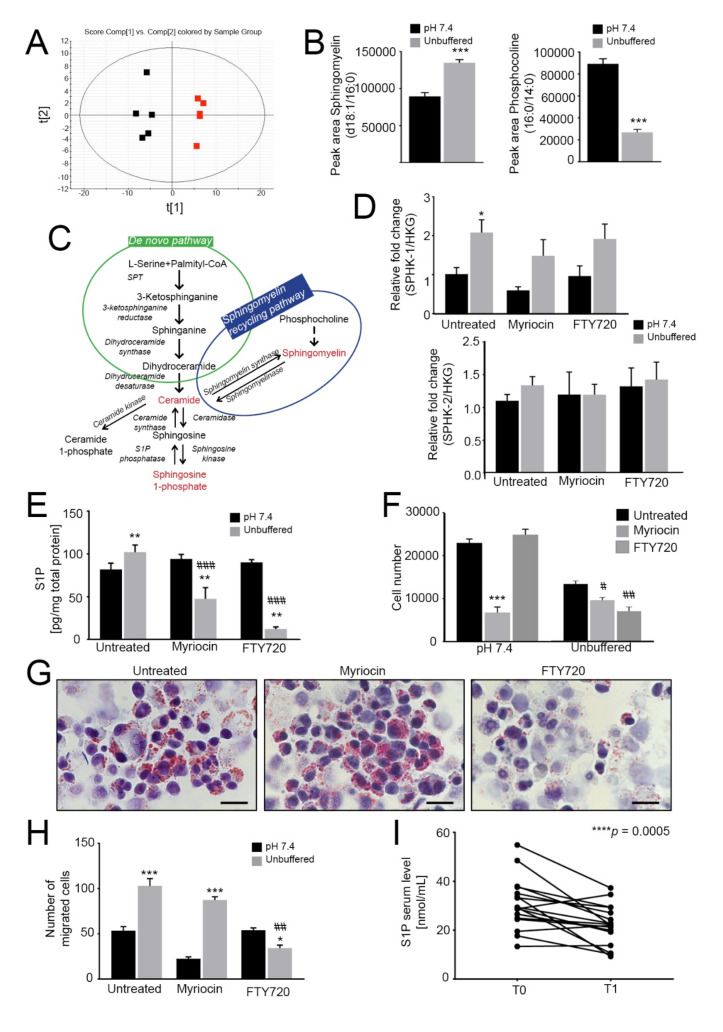
Lipidomic profiling reveals an increased sphingolipid signaling pathway. When indicated, cells were treated with myriocin at 0.75 µM or/and FTY720 at 0.15 µM. A-B. Principal component analysis of spheroids grown for 96 h under neutral or unbuffered conditions (**A**) and quantification of sphingomyelin (middle panel) and phosphocholine (right panel) in the lipidomics dataset (**B**). Data presented as mean ± S.E.M.; *n* = 5; unpaired *t*-test; *** *p* < 0.001; (**C**) Schematic representation of the sphingomyelin pathway; (**D**) Real Time PCR analysis of the indicated genes normalized to 4 housekeeping genes (HKG; GAPDH, GUSB, RNA18S, and YWAHZ) in 143B spheroids cultured for 96 h treated myriocin or FTY720. Data presented as mean ± S.E.M. of *n* = 6 RNA samples from 3 independent experiments; unpaired two-tailed Mann-Whitney test; * *p* < 0.05 for acid vs neutral; (**E**) S1P levels in 143B spheroids cultured for 96 h treated with myriocin or FTY720 were determined by ELISA. Data presented as mean ± S.E.M. of *n* = 6 samples from 3 independent experiments; unpaired *t*-test; ** *p* < 0.01 for acid versus neutral, ### *p* < 0.0001 vs untreated at the respective pH condition; (**F**) Number of 143B cells grown in spheroids under unbuffered or neutral conditions after 96 h of treatment with myriocin or FTY720. Data presented as mean ± S.E.M.; *n* = 4; unpaired *t*-test; *** *p* < 0.001 vs. neutral untreated, # *p* < 0.05 and ## *p* < 0.01 vs unbuffered untreated; (**G**) Oil Red O staining of cultured hanging-drop 143B spheroids treated as indicated and exposed to unbuffered medium for 96 h. Scale bar: 100 μm; (**H**) Migration of 143B spheroids treated as indicated. After 96 h of treatment, the spheroids were dissociated with trypsin, and the cells were allowed to migrate for 6 h in a Boyden chamber. The graph shows the quantification of cell migration. Data presented as mean ± S.E.M. of *n* = 6 samples from 3 independent experiments; unpaired two-tailed Mann-Whitney test; ## *p* < 0.01 vs untreated at the respective pH, * *p* < 0.05 and *** *p* < 0.0001 for unbuffered vs neutral; (**I**) Serum S1P levels in OS patients were determined by ELISA at the time of diagnosis (T0) and after chemotherapy (T1). *n* = 17 patients; Wilcoxon matched-paired test; *** *p* < 0.0005.

**Figure 4 cancers-13-00311-f004:**
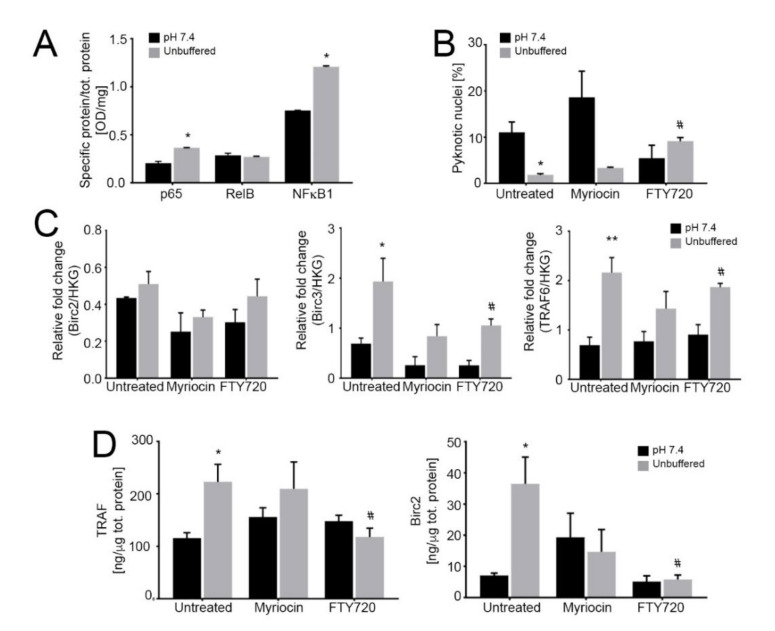
Activation of the noncanonical NF-κB pathway dramatically reduces tumor growth under acidic conditions. (**A**) Quantification of nuclear NF-κB proteins relative to total protein as an indicator of nuclear translocation and activation in 143B spheroids cultured for 96 h at pH 7.4 or in unbuffered conditions. Data are presented as the mean ± S.E.M. of *n* = 6 samples from 3 independent experiments; unpaired two-tailed Mann-Whitney test; * *p* < 0.05 for unbuffered vs neutral; (**B**) Percentage of pyknotic nuclei to total nuclei in 143B spheroids cultured for 96 h in neutral or unbuffered conditions. Data are presented as the mean ± S.E.M. of *n* = 6 samples from 3 independent experiments; unpaired two-tailed Mann-Whitney test; * *p* < 0.05 for unbuffered vs neutral, # *p* < 0.05 vs untreated at the respective pH condition; (**C**) Real Time PCR analysis of the indicated genes normalized to 4 housekeeping genes (HKG; GAPDH, GUSB, RNA18S, and YWAHZ) in 143B spheroids treated with 0.75 μM myriocin and 0.15 μM FTY720 and cultured for 96 h in neutral or unbuffered conditions. Data are presented as the mean ± S.E.M. of *n* = 6 RNA samples from 3 independent experiments; unpaired two-tailed Mann-Whitney test; * *p* < 0.05 and ** *p* < 0.01 for unbuffered vs neutral, # *p* < 0.05 vs. untreated at the respective pH condition; (**D**) ELISA quantification of TRAF and BIRC proteins expressed as ng/µg total protein in cultured 143B spheroids treated as indicated for 96 h. Data are presented as the mean ± S.E.M. of *n* = 6 replicates from 3 independent experiments; unpaired two-tailed Mann-Whitney test; * *p* < 0.05 for unbuffered vs neutral, # *p* < 0.05 vs. untreated at the respective pH condition.

**Figure 5 cancers-13-00311-f005:**
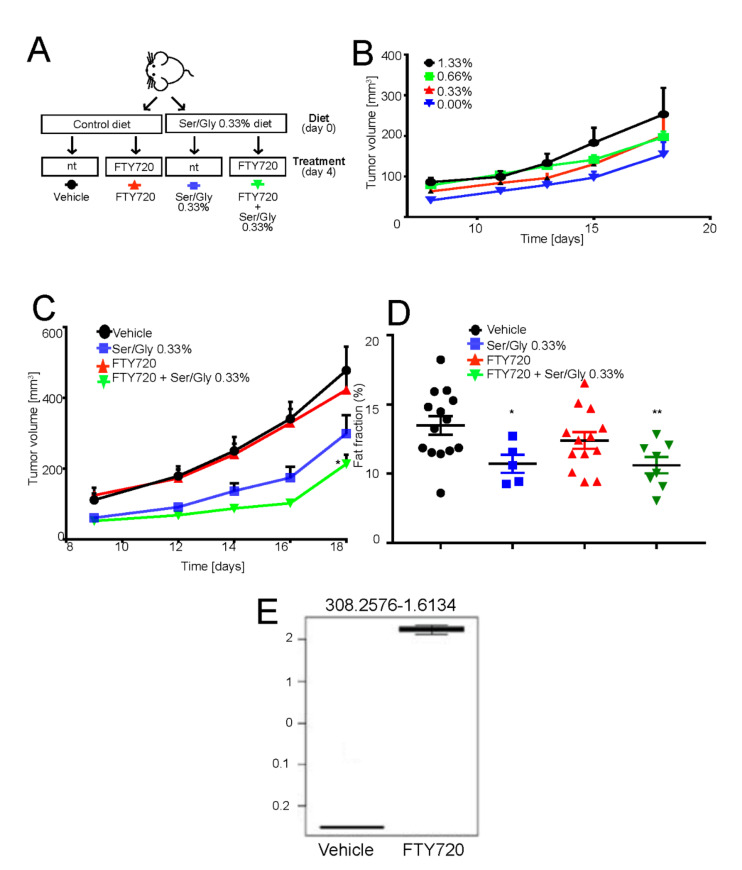
FTY720 in combination with low serine/glycine diet reduces tumor growth. (**A**) Schematic representation of the in vivo experiment. Mice were switched to normal or serine/glycine low diet at the time of subcutaneous 143B tumor implantation. Four days after injection, mice were intraperitoneally treated with 10 mg/kg of FTY720 on a 5 days on-2 days off basis. Tumor growth was monitored daily and mice were sacrificed at day 18 from injection; (**B**) Tumor volume in mice treated as on low serine/glycine diet, concentrations ranging from 0%, to 1.33%. Data are presented as the mean ± S.E.M. of *n* = 14 mice from 3 independent experiments; unpaired two-tailed Mann-Whitney test; * *p* < 0.05 vs vehicle; (**C**) Tumor volume of mice treated as in panel A. Data are presented as the mean ± S.E.M. of *n* = 14 mice from 3 independent experiments; unpaired two-tailed Mann-Whitney test; * *p* < 0.05 vs. vehicle; (**D**) Plot showing the calculated magnetic resonant imaging (MRI)-fat fraction inside the tumor region for the four investigated groups; (**E**) FTY720 quantification in tumor samples harvested 18 days from implantation, with treatment regimen described in Figure 5E, by LC-MS/MS untargeted lipidomic profiling (*n* = 7 mice).

**Table 1 cancers-13-00311-t001:** Lipid profiles of tumor xenografts. Lipid pathways majorly affected in FTY720-treated mice, significance occurring only for sphingolipid metabolism.

Lipid species	Total	Expected	Hits	Raw p	-Log(p)	Holm adjust	FDR	Impact
Sphingolipid metabolism	25	0.041545	2	0.00061373	7.396	0.049098	0.049098	0.30377
Linoleic acid metabolism	15	0.024927	1	0.024711	3.7005	1	0.98842	0
Alpha-linolenic acid metabolism	19	0.048193	1	0.047358	3.05	1	1	0

## Data Availability

The data presented in this study are available on request from the corresponding author. The data are not publicly available due to privacy and ethical reasons.

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
