# Peer review of "Exploring Metabolic Adaptations to the Acidic Microenvironment of Osteosarcoma Cells Unveils Sphingosine 1-Phosphate as a Valuable Therapeutic Target"

_cancers, 2021, doi:10.3390/cancers13020311_

Round 1

Reviewer 1 Report

“Exploring metabolic adaptations to the acid  microenvironment of osteosarcoma unveils sphingosine 1-phosphate as a valuable therapeutic target” by 6 Margherita Cortini, Andrea Armirotti, Marta Columbaro, Dario Livio Longo, Gemma Di Pompo, Elena Cannas, Alessandra Maresca, Costantino Errani, Alessandra Longhi, Alberto Righi, Valerio Carelli, Nicola Baldini, Sofia Avnet.

Authors show that the sphingolipid recycling pathway represents a survival mechanism for sarcoma cells grown under an acidic microenvironment, thus the use of a sort of inhibition of this pathway could possess therapy effect. Paper is very interesting and acquires a certain importance as new possible approach to combat osteosarcoma tumors. Few comments: a) Authors need to clarify better how lipid accumulation is linked to cell migration, in order to understand better this association; b) see lines 192-195, it is not clear to this Rev. what means sphingomyelin d18:1/16:0 (sphingomyelin usually contains only one saturated fatty acid?), as phosphocholine 16:0/14/0 (?); c) please See Fig. 3 C, we believe that Authors need to fix position of “Sphingomyelinase vs Sphingomyelin synthase; d) Table 1, please verify “alpha-linoleic acid “ vs “alpha-linolenic acid”; e) conclusion (lines 428-431) could be reconsidered and attenuated because a real structure/function data are not presented to justify the statement that a combined treatment with FTY720 and serine/glycine restrictive diet affects proliferating tumor cells and the acidic subpopulation, at the same time.

Author Response

 “Exploring metabolic adaptations to the acid  microenvironment of osteosarcoma unveils sphingosine 1-phosphate as a valuable therapeutic target” by 6 Margherita Cortini, Andrea Armirotti, Marta Columbaro, Dario Livio Longo, Gemma Di Pompo, Elena Cannas, Alessandra Maresca, Costantino Errani, Alessandra Longhi, Alberto Righi, Valerio Carelli, Nicola Baldini, Sofia Avnet.

Authors show that the sphingolipid recycling pathway represents a survival mechanism for sarcoma cells grown under an acidic microenvironment, thus the use of a sort of inhibition of this pathway could possess therapy effect. Paper is very interesting and acquires a certain importance as new possible approach to combat osteosarcoma tumors.

We thank the reviewer for having appreciated our manuscript

Few comments:

  1. Authors need to clarify better how lipid accumulation is linked to cell migration, in order to understand better this association;

We have extended the text section describing the link between migration and sphingolipid accumulation in the introduction (lines 73-79), as follows: “whereas S1P signaling inhibits apoptosis and stimulates cell proliferation and migration by directly activating signal transduction networks [23, 27] thorough five distinct receptors, S1PR1-5. The S1P-activated signal transduction pathways have distinct roles in the regulation of cancer cell proliferation, migration and/or invasion in a context-dependent manner [23]. The overall effect, is an increased metastatic potential. As an example, in colorectal cancer, the activation of the fatty acid synthase/S1P axis increases cancer cell proliferation, migration, formation of focal adhesions, and degradation of the extracellular matrix”. Hopefully, this will make things clearer.

  1. b) see lines 192-195, it is not clear to this Rev. what means sphingomyelin d18:1/16:0 (sphingomyelin usually contains only one saturated fatty acid?), as phosphocholine 16:0/14/0 (?);

Our investigation pinpointed the alterations of specific, individual lipids. As the Reviewer knows, the most common mammalian sphingolipids, including sphingomyelins, deriving from the condensation of palmitoyl-coA with serine (by SPT enzyme) share the same sphingoid base (sphingosine) and are thus indicated d18:1/. Sphingomyelin d18:1/16:0 refers to a sphingomyelin molecule, deriving from sphignosine (d18:1) and carrying a palmitic acid moiety (16:0) linked to the amino group, as amide. Further info on this particular metabolite can be found on several database, like, for example LipidMaps (https://www.lipidmaps.org/data/LMSDRecord.php?LMID=LMSP03010003). The same approach was used for phosphatidylcholines. Phosphocholine (16:0/14:0) refers to a glicerophospholipid carrying a choline headgroup and two well defined fatty acyl chains: palmitic (16:0) and myristic (14:0) acid: (https://www.lipidmaps.org/data/LMSDRecord.php?LMID=LMGP01010560). More details about the two mentioned lipids have been also added in the text of the manuscript.

  1. c) please See Fig. 3 C, we believe that Authors need to fix position of “Sphingomyelinase vsSphingomyelin synthase;

We thank the reviewer. We have corrected the figure.

  1. d) Table 1, please verify “alpha-linoleic acid “ vs “alpha-linolenic acid”;

We thank the reviewer for pointing this out. The third line of Table 1 has been modified in “alpha-linoLENIC acid metabolism”

  1. e) conclusion (lines 428-431) could be reconsidered and attenuated because a real structure/function data are not presented to justify the statement that a combined treatment with FTY720 and serine/glycine restrictive diet affects proliferating tumor cells and the acidic subpopulation, at the same time.

As suggested by the reviewer, we have cancelled the statement relative to the selective targeting of acidic subpopulation form the last paragraph of the discussion.

Reviewer 2 Report

Dear,

I have now carefully read and examined the manuscript entitled « Exploring metabolic adaptations to the acid microenvironment of osteosarcoma unveils sphingosine 1-phosphate as a valuable therapeutic target » by Cortini et al. In this study, the authors identified the activation of the sphingolipid pathway as an important process to support survival and migration of osteosarcoma (OS) cells exposed to acidic conditions. They showed that modulating levels of sphingosine 1-phosphate (S1P) through pharmacological treatment with FTY720 actually impaired acidosis-driven phenotypic adaptation in OS cells. While this study provides important new insights in the metabolic adaptation of OS cells under acidosis, this suffers from methodological flaws that must be addressed experimentally to help draw robust conclusion from the current set of data.

Major points :

1-it is unclear why the authors used 3D spheroid models to assess the effects of extracellular acidosis (vs neutral pH). Indeed, a natural gradient of acidosis (together with hypoxia) will develop within 3D spheroids suggesting that cells obtained after spheroid dissociation are phenotypically different. What is the size of the spheroids at the time of the different assays ? Besides the acidification of the extracellular medium when this is unbuffered, do the authors have any evidence for acidosis within the 3D spheroids ? LAMP2 staining ?

2-the Seahorse data and methodology are unclear. In the Y axis of Fig 2B, the data seem to be normalized per cell (even if I guess it is more « per cell density ») while in the legend, it is indicated « normalized to protein content ». Please clarify. Edit « pomol » to « pmol » in the Y axis. More importantly, there is no information about the substrates (ie glucose, glutamine, palmitate…) used during the Seahorse run. This is a crucial point since the authors conclude that LD do not support energy production in acidosis-exposed cancer cells. It is important to mention and illustrate experimentally whether or not cells may use LD as energy stores when facing a nutrient-deprived medium. Are the cells maintained in neutral and acidic conditions during the Seahorse run ?

3-The authors must discuss about the fact that acidosis-induced LD accumulation might be the consequence of cell growth arrest (as reported by Hirakawa et al Oncogene 1991 ; Delikatny et al Int J Cancer 1996 ; Barba et al Cancer Res 1999). They must also show how the growth of OS cells, in both models (2D and 3D), is changed upon incubation in neutral or acidic conditions. In material and methods section, the authors refer to « acute exposure to acidosis » but also to « chronic exposure »… Please clarify which cell models and incubation timings have been used in the different experiments.

4-In the in vivo experiment, the authors conclude that low Ser/Gly diet causes a « consistent although significant decrease in tumor volume and in the fat fraction » (lines 334-336). In Fig 5D, the decrease of fat fraction is actually statistically significant. Please clarify.

5-The material and methods section must be carefully rechecked. As already said, the cell models used in this study must be better described (acute vs chronic exposure to acidosis, which pH, which timing ?). Indicate the reference of the antibodies used in this study. What are the methods for ORO staining and measurements of mitochondria number/diameter ?

6-From my point of view, important data are in Supp Figs and it would be worth adding some of them in the main figure panels. Supp Fig 4 and 5 have several important panels which are not described in the manuscript. There is only a non-descriptive sentence in lines 123-125 which is certainly not enough for 9 figure panels ! At the first reading, we actually don’t understand why the authors refer to LC3 antibody as well as to a Lysosensor, Western blotting and immunoelectron microscopy sections. The authors must reorganize their data and edit the text accordingly.

7- A quantification of ORO-stained images would help to ascertain the differences in LD content between OS cells exposed to neutral or acidic pH conditions. This should be added in Fig 1.

Minor points :

-use « acidic » instead of « acid » as an adjective (eg « to the acidic microenvironment » in the title, lines 414, 443, 511…)

-edit the title as follows : « …microenvironment in osteosarcoma cells unveils…»

-line 55 : « advanced cancers »

-line 63 : « paracrine cross-talk between lipid-secreting cells AND ??? ». Please reword it.

-line 69 : « lipids are also oxidized to carbon dioxide… ». It is more correct to say « lipids are also oxidized to acetyl-CoA to generate energy ».

-lines 78-79 : the meaning of the sentence is unclear (redundancy between « acidic TME » and « acidic stress »). Please reword it.

-line 80 : indicate refs 31 and 32 together after « activated in TME »

-lines 90-96 : the text is in italics…

-line 92 : «indicate « Fig. 1B and Supp Fig 2 » just after « Oil RedO staining »

-in Figure 1, the use of A, B together with a and b (for inserts in the merge image) is confusing. i would suggest to use 1 and 2 for the images in panel A (instead of « a » and « b »).

-It is unclear for me at which pH the assays have been performed. Instead of « neutral » and « acid » in the different figure panels, I would suggest to indicate « pH 7.4 » and « pH 6.8 » in order to be more informative.

-line 98 « promotes intracellular lipid accumulation in OS cells »

-line 99 : « tissues from OS patients »

-line 100 : « for bright field visualization »

-line 124 : « suggest the presence of lipophagy, we could… »

-Figure 2A should be divided in two new graphs : only NDUF49 and NDUFAB1 in a new Fig 2A (or S2A) while DRP1, MFF, MFN1 and MFN2 should appear later (between current 2C and 2D panels) to follow a logical order.

-line 134 : « lipids accumulating under acidic conditions »

-line 136 : use « acyl-CoA » instead of « palmitate » term which is too restrictive here

-throughout the manuscript, symbols have been wrongly edited (see, for instance, lines 145, 153, 459…)

-line 153 : edit the text as follows « to evaluate beta-oxidation of intracellular pools of fatty acids, … »

-line 164 : the term « colocalization » is not appropriate in the figure legend. Please edit as « Immunostaining of mitochondria…. and nucleoprotein TPR in OS cells exposed to neutral or acidic pH conditions ». Please indicate that itw as done on spheroids.

-line 187 : edit as follows « Intracellular pools of sphingomyelin and S1P increase in OS cells under acidic conditions »

-Fig 3C : « sphingomyelin recycling pathway »

-throughour the manuscript and figure panels, the authors indicate « myrocin » instead of « myriocin ». Please correct.

-line 342 : « due to the FTY720 effect »

-line 387 : « secreted by tumor cells »

-line 391 : «  [71, 72]. For the future »

-line 439 « MycoAlert »… « All cell lines »

-line 479 « transferred in »

Table 2 must be referred as Supplementary Table 1 in the Supp Info pdf

Fig S3C : Y axis « number of migrating cells/field »

-In Supp Fig S3, there are no representative images for the cell migration assay, unlike what is indicated in the legend.

-Supp Fig S5 : « 7 min »

Author Response

I have now carefully read and examined the manuscript entitled « Exploring metabolic adaptations to the acid microenvironment of osteosarcoma unveils sphingosine 1-phosphate as a valuable therapeutic target » by Cortini et al. In this study, the authors identified the activation of the sphingolipid pathway as an important process to support survival and migration of osteosarcoma (OS) cells exposed to acidic conditions. They showed that modulating levels of sphingosine 1-phosphate (S1P) through pharmacological treatment with FTY720 actually impaired acidosis-driven phenotypic adaptation in OS cells. While this study provides important new insights in the metabolic adaptation of OS cells under acidosis, this suffers from methodological flaws that must be addressed experimentally to help draw robust conclusion from the current set of data.

We thank the reviewer for having read the manuscript so carefully and for having pointed out a number of important issues. We have tried to assess them in the answers below.

Major points :

1-it is unclear why the authors used 3D spheroid models to assess the effects of extracellular acidosis (vs neutral pH). Indeed, a natural gradient of acidosis (together with hypoxia) will develop within 3D spheroids suggesting that cells obtained after spheroid dissociation are phenotypically different. What is the size of the spheroids at the time of the different assays? Besides the acidification of the extracellular medium when this is unbuffered, do the authors have any evidence for acidosis within the 3D spheroids? LAMP2 staining?

We agree with the reviewer that spheroids that are cultured in long-term neutral buffered conditions (standard conditions at pH 7.4) develop a hypoxic and acidic necrotic core. However, we have performed all the experiments included in the manuscript at 96 hours, which leads the spheroids to grow, but not enough to develop a necrotic core. This is the reason why we decided to compare spheroids cultured at unbuffered conditions in respect to buffered conditions. Under unbuffered conditions, the intraspheroid pH as well as extraspheroid pH can reach the acidic value more quickly, and without the interference of a highly concentrated and artificially added buffer system (sodium bicarbonate combined to CO2). However, as suggested by the reviewer, we performed the measurement of the diameter of the spheroids to assess the spheroid growth as well as live-dead assay. Results on spheroid diameters have been also added in Supplementary Figure 3C and show that under acidity the diameter doesn’t change throughout the days, whereas at pH 7.4 it increases from 200 to 300 mm. Live/dead images of the spheroids are shown in Supplementary Figure 3D. As the reviewer will notice, few dead cells are present but they are sparse in the whole diameter of the spheroid, but at neutral or at acidic pH. Thus, 96 hours are not sufficient for the spontaneous formation of a necrotic, and possibly acidic core. As suggested by the reviewer, and as a further confirmation, we have performed Lamp2 staining of spheroids, both under neutral and acidic conditions. Indeed, under acidic conditions, the staining is very intense and is diffused to all cells in the field like the LD staining does, as expected. Under neutral conditions, instead, the staining is much lower and tends to accumulate in the regions of the spheroids that accumulate also LD (assessed by Nile Red staining). The experiment was performed at 96 hrs and we did not observe any specific localization in the spheroid necrotic core. However, the initial co-localization of LAMP2 staining and LD staining leads us to speculate that spheroids are starting to develop internal acidosis which goes along with lipid accumulation, but at a very low level compared to the spheroids cultured under unbuffered conditions (Supplementary Fig. 3C)

2-the Seahorse data and methodology are unclear. In the Y axis of Fig 2B, the data seem to be normalized per cell (even if I guess it is more « per cell density ») while in the legend, it is indicated « normalized to protein content ». Please clarify. Edit « pomol » to « pmol » in the Y axis.

As the reviewer noticed, there was a mistake in the description of the methods. The OCR experiments have been quantified by SRB measurements, i.e. by protein content. The figure has been corrected accordingly.

More importantly, there is no information about the substrates (ie glucose, glutamine, palmitate…) used during the Seahorse run. This is a crucial point since the authors conclude that LD do not support energy production in acidosis-exposed cancer cells. It is important to mention and illustrate experimentally whether or not cells may use LD as energy stores when facing a nutrient-deprived medium. Are the cells maintained in neutral and acidic conditions during the Seahorse run?

The buffer medium for the OCR readout has this composition: 1 mM pyruvate, 2 mM glutamine, and 10 mM glucose and is serum-free. The description has been added to the methods section. No palmitate or other substrates were added. This medium is a standard mito-stress medium, as the cells have no FBS supplementation, but the incubation with this nutrient-deprived medium was performed only for an hour.

Regarding the medium specifically used during the Seahorse run (1-hour incubation), all cells were maintained at pH 7.4, regardless that they had been grown at pH 7.4 or pH 6.8. Indeed, we preliminary tried to perform the assay at the 2 different pH medium and we observed an unusual trend that we believe was not due to the biological effect. We assumed that this was due to the impact of pH on the readouts. Thus, we preferred to avoid this effect by incubating the cells with acidosis until the beginning of the run, but switching to pH 7.4 during the assay. Because the run was performed at pH 7.4, we conclude that the differences in the OCR measurements are given by the different metabolic state of the cells due to the pre-incubation with medium at different pH values and not by a technical issue.

3-The authors must discuss about the fact that acidosis-induced LD accumulation might be the consequence of cell growth arrest (as reported by Hirakawa et al Oncogene 1991; Delikatny et al Int J Cancer 1996 ; Barba et al Cancer Res 1999). They must also show how the growth of OS cells, in both models (2D and 3D), is changed upon incubation in neutral or acidic conditions. In material and methods section, the authors refer to « acute exposure to acidosis » but also to « chronic exposure »… Please clarify which cell models and incubation timings have been used in the different experiments.

We have added the references suggested by the reviewer, along with a brief discussion in lines 409-412.

We have extended the material and methods section with the descriptions of acute and chronic exposures and these have also been added to the figure legends. Hopefully this will clarify things.

A full growth curve, in acid and neutral is shown in Figure 3F. Data on the growth of OS cells under different pH values, in 2D, has been also previously published (Avnet et al. DOI: 10.18632/oncotarget.11503).

4-In the in vivo experiment, the authors conclude that low Ser/Gly diet causes a « consistent although significant decrease in tumor volume and in the fat fraction » (lines 334-336). In Fig 5D, the decrease of fat fraction is actually statistically significant. Please clarify.

We thank the reviewer for letting us notice the typo. We have rephrased the sentence and highlighted that the difference seen with the ser/gly diet alone is lower than that seen with the combination treatment with FTY720.

5-The material and methods section must be carefully rechecked. As already said, the cell models used in this study must be better described (acute vs chronic exposure to acidosis, which pH, which timing?). Indicate the reference of the antibodies used in this study. What are the methods for ORO staining and measurements of mitochondria number/diameter?

We thank the reviewer for noticing that the ORO section was missing. It has been now added to the materials and methods section, as well as the description of the mitochondria diameter and number. The description of the chronic vs acute exposure of the cells has been extended. Hopefully it is now more clear. Moreover, the figure legends have been added of the descriptions of the timing of each experiment and the conditions used.

The references of the antibodies have been added in the Materials and Methods section.

6-From my point of view, important data are in Supp Figs and it would be worth adding some of them in the main figure panels. Supp Fig 4 and 5 have several important panels which are not described in the manuscript. There is only a non-descriptive sentence in lines 123-125 which is certainly not enough for 9 figure panels! At the first reading, we actually don’t understand why the authors refer to LC3 antibody as well as to a Lysosensor, Western blotting and immunoelectron microscopy sections. The authors must reorganize their data and edit the text accordingly.

We thank the reviewer for the suggestion. We have included in the main figures, a panel showing that autophagy does not increase under acidic conditions and a panel showing the presence of lipophagic membranes (Figure 1 F, G, H). We have also extended the description of the figures referring to autophagy in the main text and in the discussion.

7- A quantification of ORO-stained images would help to ascertain the differences in LD content between OS cells exposed to neutral or acidic pH conditions. This should be added in Fig 1.

We agree with the reviewer. Thanks for the suggestion. We have added a quantification of the ORO staining, quantified as um2 of LD area/cell, for neutral vs acidic conditions in Figure 1E, as suggested.

Minor points :

-use « acidic » instead of « acid » as an adjective (eg « to the acidic microenvironment » in the title, lines 414, 443, 511…)

The text has been adjusted accordingly.

-edit the title as follows : « …microenvironment in osteosarcoma cells unveils…»

The text has been adjusted accordingly.

-line 55 : « advanced cancers »

The text has been adjusted accordingly.

-line 63 : « paracrine cross-talk between lipid-secreting cells AND ??? ». Please reword it.

The text has been corrected as suggested.

-line 69 : « lipids are also oxidized to carbon dioxide… ». It is more correct to say « lipids are also oxidized to acetyl-CoA to generate energy ».

The text has been adjusted accordingly.

-lines 78-79 : the meaning of the sentence is unclear (redundancy between « acidic TME » and « acidic stress »). Please reword it.

The text has been adjusted accordingly.

-line 80 : indicate refs 31 and 32 together after « activated in TME »

The text has been adjusted accordingly.

-lines 90-96 : the text is in italics…

The text has been adjusted accordingly.

-line 92 : «indicate « Fig. 1B and Supp Fig 2 » just after « Oil RedO staining »

The text has been adjusted accordingly.

-in Figure 1, the use of A, B together with a and b (for inserts in the merge image) is confusing. i would suggest to use 1 and 2 for the images in panel A (instead of « a » and « b »).

The figure has been adjusted as suggested.

-It is unclear for me at which pH the assays have been performed. Instead of « neutral » and « acid » in the different figure panels, I would suggest to indicate « pH 7.4 » and « pH 6.8 » in order to be more informative.

All the assays in the 3D model have been performed in unbuffered conditions, which means that the spheroids adjust the pH to their own demand and growth rate. We have measured the pH several times and the values are reproducibly between 6.6 and 6.7. All the experiments in the 2D model have instead been performed at pH 6.8, because monolayer cells tend to acidify more than 3D spheroids in unbuffered conditions.

We have taken into account the suggestion of the reviewer and now refer to “unbuffered” for all the experiments carried out in 3D and “pH 6.8” for the experiments in 2D.

-line 98 « promotes intracellular lipid accumulation in OS cells »

The text has been adjusted accordingly.

-line 99 : « tissues from OS patients »

The text has been adjusted accordingly.

-line 100 : « for bright field visualization »

The text has been adjusted accordingly.

-line 124 : « suggest the presence of lipophagy, we could… »

The text has been adjusted accordingly.

-Figure 2A should be divided in two new graphs : only NDUF49 and NDUFAB1 in a new Fig 2A (or S2A) while DRP1, MFF, MFN1 and MFN2 should appear later (between current 2C and 2D panels) to follow a logical order.

We have split the graph in two, as suggested by the reviewer.

-line 134 : « lipids accumulating under acidic conditions »

The text has been adjusted accordingly.

-line 136 : use « acyl-CoA » instead of « palmitate » term which is too restrictive here

The text has been adjusted accordingly.

-throughout the manuscript, symbols have been wrongly edited (see, for instance, lines 145, 153, 459…) The text has been adjusted accordingly.

-line 153 : edit the text as follows « to evaluate beta-oxidation of intracellular pools of fatty acids, … »

The text has been adjusted accordingly.

-line 164 : the term « colocalization » is not appropriate in the figure legend. Please edit as « Immunostaining of mitochondria…. and nucleoprotein TPR in OS cells exposed to neutral or acidic pH conditions ». Please indicate that itw as done on spheroids.

The text has been adjusted accordingly.

-line 187 : edit as follows « Intracellular pools of sphingomyelin and S1P increase in OS cells under acidic conditions »

The text has been adjusted accordingly.

-Fig 3C : « sphingomyelin recycling pathway ».

Typo has been corrected.

-throughour the manuscript and figure panels, the authors indicate « myrocin » instead of « myriocin ». Please correct. We have corrected the typos.

-line 342 : « due to the FTY720 effect »

The text has been adjusted accordingly.

-line 387 : « secreted by tumor cells »

The text has been adjusted accordingly.

-line 391 : «  [71, 72]. For the future »

The text has been adjusted accordingly.

-line 439 « MycoAlert »… « All cell lines »

The text has been adjusted accordingly.

-line 479 « transferred in » The text has been adjusted accordingly.

Table 2 must be referred as Supplementary Table 1 in the Supp Info pdf.

The table is now referred as Supplementary Table 1.

Fig S3C : Y axis « number of migrating cells/field »

The test has been corrected accordingly.

-In Supp Fig S3, there are no representative images for the cell migration assay, unlike what is indicated in the legend.

The figure legend has been corrected.

-Supp Fig S5 : « 7 min »

The figure has been corrected.

Round 2

Reviewer 2 Report

In the revised version of their manuscript, the authors have now addressed my initial comments. In particular, they have better explained the cell models used in the study and provided enough methodological details in the methods section. They have also improved the overall quality of the text and the figures.

Minor points:

-line 74: "through"

-line 76: remove the coma after "the overall effect"

-line 236: "sphingosine"

line 238: "glycerophospholipid"

In panels from figures 3 and 4, "myriocin" is still misspelled.

Author Response

In the revised version of their manuscript, the authors have now addressed my initial comments. In particular, they have better explained the cell models used in the study and provided enough methodological details in the methods section. They have also improved the overall quality of the text and the figures.

Minor points:

-line 74: "through" Corrected

-line 76: remove the coma after "the overall effect" Corrected

-line 236: "sphingosine" Corrected

line 238: "glycerophospholipid" Corrected

In panels from figures 3 and 4, "myriocin" is still misspelled. Corrected